Automated identification of insect vectors of Chagas disease in Brazil and Mexico: the Virtual Vector Lab

Gurgel-Gonçalves Rodrigo 1
Komp Ed 2
Campbell Lindsay P. 3
Khalighifar Ali 3
Mellenbruch Jarrett 4
Mendonça Vagner José 1 5
Owens Hannah L. 3 6
de la Cruz Felix Keynes 7
Peterson A Townsend town@ku.edu 3
Ramsey Janine M. 7
1 Faculty of Medicine, Universidade de Brasília , Brasilia , DF , Brazil
2 Information and Telecommunication Technology Center, University of Kansas , Lawrence , KS , United States
3 Biodiversity Institute, University of Kansas , Lawrence , KS , United States
4 Spencer Art Museum, University of Kansas , Lawrence , KS , United States
5 Centro de Ciências da Saúde, Universidade Federal do Piauí , Brazil
6 Florida Museum of Natural History, University of Florida , Gainesville , FL , United States
7 Centro Regional de Investigación en Salud Pública, Instituto Nacional de Salud Publica , Tapachula , Chiapas , Mexico
Oppert Brenda
Electronic publication date: 2017 Apr 18
Publication date: 2017
Volume: 5
Electronic Location ID: e3040
Received 2016 Sep 26; Accepted 2017 Jan 28
Copyright: ©2017 Gurgel-Gonçalves et al.
Copyright year: 2017
Copyright holder: Gurgel-Gonçalves et al.
License: This is an open access article distributed under the terms of the Creative Commons Attribution License, which permits unrestricted use, distribution, reproduction and adaptation in any medium and for any purpose provided that it is properly attributed. For attribution, the original author(s), title, publication source (PeerJ) and either DOI or URL of the article must be cited.
License URL: https://creativecommons.org/licenses/by/4.0/

Keywords: Identification, Chagas disease, Triatominae, Automation, Primary occurrence data

Funding: CONACyT FONSEC #161405 We thank the Office of the Provost, of the University of Kansas, for their support of this initiative. This project also was supported by CONACyT FONSEC #161405 to JMR. The funders had no role in study design, data collection and analysis, decision to publish, or preparation of the manuscript.

==============================
Identification of arthropods important in disease transmission is a crucial, yet difficult, task that can demand considerable training and experience. An important case in point is that of the 150+ species of Triatominae, vectors of Trypanosoma cruzi, causative agent of Chagas disease across the Americas. We present a fully automated system that is able to identify triatomine bugs from Mexico and Brazil with an accuracy consistently above 80%, and with considerable potential for further improvement. The system processes digital photographs from a photo apparatus into landmarks, and uses ratios of measurements among those landmarks, as well as (in a preliminary exploration) two measurements that approximate aspects of coloration, as the basis for classification. This project has thus produced a working prototype that achieves reasonably robust correct identification rates, although many more developments can and will be added, and—more broadly—the project illustrates the value of multidisciplinary collaborations in resolving difficult and complex challenges.

Introduction

The challenge of identifying insects important in public health or agriculture is significant, since efficient surveillance and mitigation often depend on non-expert or community participation, and yet often must focus on particular species (e.g., Dias, Silveira & Schofield, 2002). Dichotomous keys have long been the primary tool for most taxonomic identifications, although their use is limited by the expertise required (Gaston, 1992; Drew, 2011). Alternatives, such as creation of cooperative extension programs, as at state and federal land-grant institutions across the United States (Allen & Rajotte, 1990), are expensive in terms of resources and personnel; recent approaches creating online field guides have made identification more efficient, but again are labor-intensive (Stevenson, Haber & Morris, 2003). DNA barcoding has also been explored as a substitute for morphology-based identification, but its cost and lack of efficacy preclude its use outside research or emergency efforts (Hebert & Gregory, 2005; Meier et al., 2006).

The possibility of automating the identification process has long been discussed and explored (Weeks et al., 1997; Gaston & O’Neill, 2004; MacLeod, 2007; MacLeod et al., 2010). Progress has been made in using acoustic or other waveforms to identify some taxa (e.g., birds, Orthoptera; Moore & Miller, 2002; Chesmore, 2004; Acevedo et al., 2009), and the Digital Automated Identification SYstem (DAISY) has been used with some success based on optical imagery in several insect groups (Weeks et al., 1997; Weeks et al., 1999; Watson, O’Neill & Kitching, 2003). Certain contributions have focused on improving the algorithms and approaches to identification (Schroder et al., 1995; Weeks et al., 1997; Kang, Jeon & Lee, 2012; Kang, Song & Lee, 2012; Wang et al., 2012a; Wang et al., 2012b), whereas others have made steps toward more full automation of the process (Arbuckle et al., 2001; Mayo & Watson, 2007; Yang et al., 2015).

Progress in automating identification of medically important vectors, however, has been slower. Although some advances have been made toward automated identification of mosquitoes (Culicidae) using wing venation (Zhou, Ling & Rohlf, 1985; Lorenz, Ferraudo & Suesdek, 2015), more effort has centered on DNA barcoding as a solution (Cywinska, Hunter & Hebert, 2006; MBI, 2009). For sand flies (Psychodidae), the only steps towards automating identification have been via mass-spectrophotometry of proteins (Mathis et al., 2015) and DNA barcoding (Pinto et al., 2015). Automated identification of ticks (Ixodidae and Argasidae) and kissing bugs (Triatominae) has not been attempted, although many morphometric studies have been developed (e.g., Voltsit & Pavlinov, 1995; Dujardin & Slice, 2007), and online identification keys have now been implemented to provide some level of identification support at least in Brazil (http://triatokey.cpqrr.fiocruz.br/).

Here, we present a fully-automated visual identification system for kissing bugs (Triatominae: Hemiptera: Reduviidae; full nomenclatural details for each species are provided in Tables 1 and 2), vectors of Trypanosoma cruzi, the etiologic agent of Chagas disease. Beginning with a high-quality photographic image, we have automated all steps from preparing and processing the image to identifying landmarks and running and interpreting identification routines. We implement this system for the triatomines of Brazil (39 species) and Mexico (12 species, including the three haplogroups of the Triatoma dimidiata complex), but are already broadening the scope of the project to all triatomines (https://vectorlab.org), and are exploring the transfer of the techniques employed to other medically important arthropod groups.

Table 1 Summary of species analyzed, sample sizes of photographs, and identification success rates, for the 39 species of Brazilian triatomine bugs analyzed in this study.

Species	Sample size	Success rate	
Cavernicola lenti Barrett & Arias, 1985	15	93.3	
Eratyrus mucronatus Stål, 1859	10	80.0	
Panstrongylus diasi Pinto & Lent, 1946	30	96.7	
Panstrongylus geniculatus (Latreille, 1811)	45	93.3	
Panstrongylus lignarius (Walker, 1873)	28	85.7	
Panstrongylus lutzi Neiva & Pinto, 1923	34	88.2	
Panstrongylus megistus Burmeister, 1835	84	91.7	
Psammolestes tertius Lent & Jurberg, 1965	29	100.0	
Rhodnius brethesi Matta, 1919	28	96.4	
Rhodnius domesticus Neiva & Pinto, 1923	27	96.3	
Rhodnius milesi Carcavallo, Rocha, Galvão & Jurberg, 2001	37	89.2	
Rhodnius montenegrensis Rosa et al. 2012	39	84.6	
Rhodnius nasutus Stål, 1859	73	82.2	
Rhodnius neglectus Lent, 1954	60	83.3	
Rhodnius pictipes Stål, 1872	43	95.3	
Triatoma arthurneivai Lent & Martins, 1940	32	78.1	
Triatoma baratai Carcavallo & Jurberg, 2000	29	82.8	
Triatoma brasiliensis Neiva, 1911	64	76.6	
Triatoma carcavalloi Jurberg, Rocha & Lent, 1998	38	86.8	
Triatoma circummaculata (Stål, 1859)	21	85.7	
Triatoma costalimai Verano & Galvão, 1958	63	85.7	
Triatoma delpontei Romana & Abalos, 1947	29	86.7	
Triatoma guazu Lent & Wygodzinsky, 1979	28	64.3	
Triatoma infestans (Klug, 1834)	54	83.3	
Triatoma juazeirensis Costa & Felix, 2007	21	81.0	
Triatoma lenti Sherlock & Serafim, 1967	19	78.9	
Triatoma maculata (Erichson, 1848)	39	89.7	
Triatoma matogrossensis Leite & Barbosa, 1953	32	75.0	
Triatoma melanica Neiva & Lent, 1941	29	79.3	
Triatoma pintodiasi Jurberg, Cunha & Rocha, 2013	25	88.0	
Triatoma platensis Neiva, 1913	27	74.1	
Triatoma pseudomaculata Correa & Espínola, 1964	55	70.9	
Triatoma rubrovaria (Blanchard, 1843)	54	59.3	
Triatoma sherlocki Papa, Jurberg, Carcavallo, Cerqueira & Barata, 2002	31	93.5	
Triatoma sordida (Stål, 1859)	96	81.2	
Triatoma tibiamaculata (Pinto, 1926)	41	92.7	
Triatoma vandae Carcavallo, Jurberg, Rocha, Galvão, Noireau & Lent, 2002	29	69.0	
Triatoma vitticeps (Stål, 1859)	47	85.1	
Triatoma williami Galvão, Souza & Lima, 1965	17	70.6	

Table 2 Summary of species analyzed, sample sizes of photographs, and identification success rates, for the 12 species of Mexican triatomine bugs analyzed in this study.

Species	Sample size	Success rate	
Panstrongylus rufotuberculatus (Champion, 1899)	7	100.0	
Triatoma barberi Usinger, 1939	29	72.4	
Triatoma dimidiata (Latreille, 1811) HG1	44	70.5	
Triatoma dimidiata (Latreille, 1811) HG2	30	76.7	
Triatoma dimidiata (Latreille, 1811) HG3	40	82.5	
Triatoma gerstaeckeri (Stål, 1859)	12	83.3	
Triatoma longipennis Usinger, 1939	51	72.5	
Triatoma mazzottii Usinger 1941	22	77.3	
Triatoma mexicana (Herrich-Schaeffer, 1848)	45	80.0	
Triatoma nitida Usinger, 1939	15	46.7	
Triatoma pallidipennis Stål, 1872	43	90.7	
Triatoma phyllosoma (Burmeister, 1835)	58	46.6	

The Automatization Workflow

Photographs for input

We observed early in the course of this project that the triatomine photographs available to us were of variable and generally low quality. As a consequence, we set about designing an apparatus that would permit taking consistent, high-quality, repeatable photographs of triatomines at low cost (Fig. 1). It is constructed of acrylic polymer and alumina trihydrate (commercially available as Corian®), with metal fittings and a metal support rod. The upper platform of the apparatus has a cradle in which an iPod touch® (or cell phone) is nestled, as well as a ring of LED lights to illuminate the bug for photography. The lower platform includes a size standard, as well as a pin point on which the specimen can be affixed by impaling through the abdomen, such that the pin is not visible in the resulting dorsal-view image. The lower platform is an even blue color, which is an important feature for image processing (see below).

Figure 1 Photographs of the apparatus designed for capture of high-quality images of triatomine bugs for this project.

(A) and (B) show a view from above; (C) and (D) show a lateral view, with the lighting ring, and an insect (not a triatomine) impaled on the pin (note that the pin does not protrude through the dorsum and thus is not visible in the image).

The iPod used for photography is equipped with an Olloclip 7× macro lens to optimize close-range photography. The 5th-generation iPod generates 5 MP images, whereas the 6th-generation iPod delivers 8 MP images. Photos are then uploaded to a DropBox folder for communication to the processing and identification modules, a step that is soon to be automated so that contribution of photos to the project is automatic. The iPod includes videos in Portuguese (https://vimeo.com/187498921/e4ef52cd46), Spanish, and English, showing how to use the apparatus. In future iterations of development of these functionalities, we will take advantage of the GPS geotagging functionality of the iPod to add geographic information automatically to photos and data records, although care must be taken either to match the geographic coordinates of the photo with the locality where the specimen was collected or to provide an independent georeference.

The apparatus was used to assemble a large reference collection of photos for as many triatomine species that occur in Brazil and Mexico as were available for testing these approaches. Triatomines were photographed from collections from across Brazil (Universidade de Brasília, FIOCRUZ Bahia, FIOCRUZ Rio de Janeiro, FIOCRUZ Minas Gerais, Universidade Estadual Paulista) and Mexico (Centro Regional de Investigación en Salud, Instituto Nacional de Salud Pública México; Laboratorio Estatal de Salud Pública de Guanajuato; Universidad Autónoma Benito Juárez, Oaxaca; Universidad Autónoma de Nuevo León, Monterrey). Only by the kind collaboration of colleagues at these institutions was it feasible to assemble the set of photographs necessary for this work.

Our protocol for capturing reference images, which also represents best practices for unknown bugs to be identified, is as follows. If bugs are preserved in ethanol, they are allowed to dry for 10 min; if dried, specimen pins and labels were removed whenever possible to reduce bias in color measures of the dorsum. If specimens are from insectaries (generally no more than two generations of captive propagation) or recent field collections, they are killed by freezing for 15 min, cleaned with a damp brush, allowed to dry for a few minutes, and placed on the pin point in the photo apparatus. When possible (i.e., the specimen is not needed as a permanent specimen voucher), the first pair of legs is removed to simplify image processing. The specimen is photographed with the following steps: (1) size standard magnet positioned next to the specimen with its upper surface flush with the specimen’s dorsum (back); (2) LED light ring turned on, along with the camera application on the iPod, 7× magnification lens attached; (3) specimen centered in the picture frame, attempting to fill >90% of the field with the specimen’s length; (4) lens focused and light meter reading checked; (5) photo taken and quality checked on the iPod; (6) images uploaded to DropBox and quality checked again. Photography takes ∼1 min per specimen.

Image preparation and processing

The photographs produced using the apparatus described above are quite consistent in orientation, resolution, and quality, which enables full automation of their processing. The sequence is illustrated in Fig. 2: major steps include removing (digitally) background and extraneous body parts (legs, antennae), orientation and identification of landmarks, measurements and calculation of ratios, and submission to identification routines. Descriptions of our processing steps follow.

Figure 2 Summary of major steps in the processing of an example image of an individual of Triatoma brasiliensis.

(A) raw image, (B) background removed to create a binary image, (C) legs and antennae removed and edge identified, (D) insect body filled and landmarks added, and (E) final image with landmarks overlaid.

Correct image and extract the core body from the raw image

A first step in processing images is to correct the image for lens distortion. The macro lens attachment to the iPod introduces significant “pincushion” distortion to images. Since all photos are taken with the same lens arrangement, and given only minor variation in the distance from lens to target, a correction is applied using a fixed set of parameters in the plug-in lens-distortion module of GIMP, an open-source Photoshop-like application. The two parameters set are main = 12.0, which specifies how much spherical correction to introduce, and edge = 6.0, which indicates the amount of additional spherical correction at image edges.

A second step is to remove the background from the image. The photo apparatus includes a baseplate with a solid blue color, which greatly facilitates isolating the sample in the photograph. However, significant variation in the background color exists among and even within images, owing to variation in ambient lighting, shading, etc. This effect is particularly noticeable and critical near the edge of the bug’s body. Hence, background pixels are identified not as specific RGB colors, but by a relation among RGB values, as follows: for each of the three color dimensions (R, G, and B), a pixel is considered as background B > 0.75 and R < 0.7B and G < 0.9B, or if B > 0.65 and (R + G) < 1.25B and 1.5R < G. The first condition is that which most often is applicable (i.e., blue is relatively large and dominates red and green, to avoid including other bright colors or white; particular values were chosen empirically, based on numerous trials).

A third step identifies the specimen’s body edge. An edge trace is extracted that satisfies several properties: (1) the edge is divided into left and right portions; (2) the edge is continuous such that each edge pixel is adjacent to two other edge pixels; (3) the edge is exactly one pixel wide such that for any edge pixel, exactly two of its eight neighbors are also part of the edge; (4) the edge progresses monotonically in the y-direction (i.e., sagittally). These restrictions make for efficient operations, such as following edges, detecting changes in direction, or identifying notable features. If one draws a horizontal line across the specimen image, it intersects the edge at exactly two points: one on the left side and one on the right side of the body.

The fourth step is to clip the legs and antennae from the image. Appendages appear as relatively narrow, straight segments. Detecting such segments is a two-step process. First, the binary image is traversed row-by-row and then column-by-column, recording narrow bands of bug as sequences of zeroes (0) of length less than the maximum segment width. Adjacent bands that form segments are collected for removal. Because this procedure is applied first in the horizontal direction, and then in the vertical direction, removal is best for appendages aligned horizontally or vertically; it can handle appendages oriented at intermediate angles, so long as the width of the horizontal or vertical slice is less than the maximum width parameter.

Finally, triangles left by previous clipping operations are smoothed. The appendages removed in the previous step are sliced off along a vertical or horizontal line where the appendage attaches to the body. If the edge at the attachment point is not exactly horizontal or vertical, an artificial triangular shape is generated along the edge. A pass along the edge identifies these triangular shapes as 90° changes in direction; when such changes are detected, the two adjoining horizontal and vertical segments are replaced by the hypotenuse connecting their endpoints.

Detect landmarks on body edge

A specialized detection function was developed for each landmark, as is described in detail below, in general not depending on detection of other landmarks; left/right landmarks (e.g., left eye center and right eye center) are detected as pairs. These algorithms assume an approximate vertical line of symmetry, and use this assumption to improve detection. Comparisons with other landmark positions are used in validation steps, which may cause iteration of certain functions using additional constraining arguments.

To locate the anterior and posterior extrema (clypeus and final tergits or wing tips, respectively) of the dorsal view of each specimen, the points along the edge with maximum and minimum y-values are identified. The point at the anterior extreme of the head (clypeus) relies on correct edge extraction at the head and neck juncture, and correct antenna exclusion. If the wings extend beyond the final tergits, then the point at the posterior extreme of the specimen will be beyond the end of the abdomen. We note that distinguishing between wing tips and final tergits as comprising the posterior extreme of the specimen is a challenge for future development of this tool, and may reduce measurement variation and improve identification in very useful ways.

The eye center and radius are identified next. The eyes are the most distinctive features of the bug, appearing as hemispheres; no other portion of the edge has this shape. The Circle Hough Transform is a well-known technique in digital image processing for detecting circles of arbitrary radius (Ballard, 1981). However, for our purposes, to reduce processing time and avoid false positives, a valid range of eye radius values for each image is identified. First, the expected eye radius is estimated, and the search is restricted to hemispheres with radii close to this estimate. An estimate of the eye radius is core body length/48 (estimated empirically); hence circles with radius in the range of the estimate ±10 pixels are sought. Providing a radius range parameter to the transform significantly reduces execution time, and avoids many false positives.

The Hough Transform returns many possible matches, but left and right hemispheres that constitute a reasonable hypothesis of the pair of eyes for the bug are the target. As such, in a series of conditionals and empirically-tuned exploratory filters, the following conditions are imposed: (1) left and right eyes should be in approximately the same position vertically; (2) the two eyes should have nearly the same radius; and (3) the eye centers should be close to one another. An internal scoring scheme identifies the pair of putative eyes with the highest score, but also requires a minimum match score. If no pair scores above this minimum, the detector returns “NotFound,” rather than values that are likely incorrect. This approach detects left and right eye pairs, but can be confused if one eye is obscured in an image, in which case the routine fails to detect either eye. In future versions, we hope to develop single-eye detectors to provide at least some useful information from lower-quality images or images of damaged individual specimens.

The next landmark to be located is the antenna junction. To this end, the routine searches below the anterior extremus of the specimen, which typically is a relatively straight vertical line segment, seeking the first significant horizontal deviation in direction. This position is compared to the eye position on the body side: if it appears at or below the eye, the location is rejected. A secondary search routine uses the position of the antenna junction on the opposite body side for guidance.

A major challenge in locating landmarks focuses on locating the pronotum fore lobe, humerus, and midpoints. Because of the restriction that the extracted edge increases monotonically in the y dimension, the anterior-most tips of the pronotum fore lobe may be clipped. Given the extracted edge, the anterior lateral margin of the pronotum fore lobe appears as a dramatic directional change along the edge. Along the collar, the edge is quite horizontal, but it turns downward at a relatively steep and constant angle at the margin, which is the focus of our detection efforts. The posterior lateral margin (humerus) of the pronotum can be identified similarly: the humerus widens at a relatively constant angle until it attains its maximum width, at which point the edge angle changes, again allowing detection. These paired edge points are used to detect midpoints between left and right sides. Finally, maximum body width is extracted by simply scanning the core body image row by row to find the maximum difference between the right and left edges; the accuracy of this measure depends on the accuracy with which the legs have been clipped off in earlier steps.

Other features that used in our analyses are extracted and characterized as follows. Body area is summarized as the area of the head, pronotum, and abdomen, comprising the entire white area in Fig. 2D. The head area is the portion of the body core anterior to the pronotum fore lobe. The pronotum area (approximate) is the portion between the fore lobe and humerus angle landmark pairs. Abdomen area is derived by subtraction.

Classify sample

The landmarks described above are then turned into a set of descriptors to be used in the identification exercises. Of course, it is possible to compute large numbers of measurements for these uses: distances among landmarks, angles between different line segments connecting landmarks, ratios of pairs of measurements, etc. However, because more is not always better, a first step was to select a small, but maximally effective, set of features with which to distinguish among the triatomine species.

Three criteria are employed in the process of choosing the set of descriptors to a core suite for identification: (1) The average of each pair of symmetric measurements is used as a single feature, which reduces minor, random error in automatic detection of landmark locations. (2) Taxonomic research on triatomines was reviewed to detect and quantify salient and particularly relevant features. (3) Extensive exploratory testing was done with different combinations of the candidate features.

All measurements are made in terms of pixel units on the digital image; for size-related measurements explored later in this study, we use size-standard information to convert to absolute measurements (mm). To avoid problems of scaling, and of size variation related to individual age, a series of ratios of pairs of measurements is used, such that classification features are unitless. The features chosen were thus as follows: total length/clypeus—pronotum forelobe midpoint, total length/mean lateral eye margin, total length/mean eye center, total length/mean lateral pronotum forelobe, total length/pronotum forelobe midpoint to pronotum-humeral angle midpoint, total length/mean lateral pronotum humeral angle, total length/total area, total length/maximum body width, and total length/mean eye center—mean lateral pronotum fore lobe.

To explore the utility of interior features (i.e., within the outline of the body), and more in particular the possible utility of characteristics of coloration, we examined one species pair that proved problematic to identify in initial analyses: T. pallidipennis and T. phyllosoma. The triatomine experts on the project team indicated that these two species are easily differentiated by the color of the corium: in T. pallidipennis, it is mostly white, whereas in T. phyllosoma, it is black with two light spots. Hence, a rectangle bounded by the pronotum humerus edges (which are already landmarked), extending posteriorly twice the length of the pronotum is extracted. Within the rectangle, a triangular area roughly corresponding to the scutellum is identified. Finally, the ratio of the average gray-scale value within the rectangle and outside the triangle to the average grayscale value inside the triangle is calculated; this ratio is used as a further characteristic of the bugs that could be included in identification exercises. This ratio is quite high in T. pallidipennis, and low in T. phyllosoma.

While working on color variation around the scutellum, we noticed pattern variation on the pronotum among various species. For many species, the pronotum has a relatively constant color, but T. brasiliensis shows a series of stripes. For T. pintodiasi and T. carcavalloi, the upper portion of the pronotum is relatively dark, and the lower portion is much brighter. Instead of attempting to describe the actual patterns, and developing detection algorithms for each, we explored a very simple summary measurement. We record the color variation among the pixels of the pronotum.

Finally, based on knowledge on the part of triatomine specialists on the team, we perceived the utility of exploring measures of absolute size. As a step to this end, we used total length, but translated into absolute measurements (mm) via reference to the size of the size standard (which was detected and measured using routines for detecting circular structures described above). This single absolute-size measure proved useful in improving classification of some difficult species pairs.

A key observation from initial data analyses was the improvement of identifier performance using smaller pools of species from among which the classifier chooses. The use of pairwise combinations of binary classifiers for multi-class problems is well-described in the literature (Wu, Lin & Weng, 2004). For classification, our routines use the nnet package in the language R (https://cran.r-project.org/web/packages/nnet/) which implements a feed-forward neural network model (Ripley, 1996) These routines require few parameters, relying instead almost exclusively on the data provided for model calibration. To construct binary classifiers for every pair of species, among 39 species from Brazil (39 × 38)∕2 = 741 and 12 species from Mexico (12 × 11)∕2 = 66 such classifiers exist, and a simple voting scheme is used to combine results, such that the species name chosen by the most pairwise classifiers is selected. This pairwise combination of binary classifiers is more effective than a single multi-class classifier, particularly when the target pool is restricted to those species represented in the fauna predicted by ecological niche models to be present at that site (see next section).

Identify faunal subsets based on modeled distributions

To reduce suites of species under comparison (see below), it was necessary to summarize the geographic distributions of the 39 Brazilian and separately the 12 Mexican triatomine species of interest in our identification efforts. Using ecological niche modeling techniques and data sources described in detail elsewhere (Gurgel-Gonçalves et al., 2012; Ramsey et al., 2015), we created binary maps summarizing potential geographic distributions for each of the 39 Brazilian and 12 Mexican species. Ecological niche models were drawn from our previous publications (Gurgel-Gonçalves et al., 2012; Ramsey et al., 2015) for 14 species. For 17 more species not included in the earlier publications, we developed new ecological niche models following the same protocols as in those publications. We converted them to binary maps using an adjusted least training presence threshold approach, in which we sought the highest cut-off that included (100–E)% of the calibration data, where E is the percentage of occurrence data likely to include meaningful errors (Peterson, Papeş & Soberón, 2008) and was set at 5%. For the remaining 20 species, for which <10 occurrence points were available, and robust model creation using the methods of the previous publication were not possible; we instead used a simple known-extent-of-occurrence approach to delineate distributional areas of potential distributions. A 20 km buffer was created around the known occurrence points of each species, and considered all sites within that buffer as within the distributional area for the species.

Resulting binary distribution maps of each species were stacked using the raster package in R (Hijmans, 2015). Values within each grid cell were extracted to create a presence/absence matrix (PAM) with columns representing individual species and rows representing all 5 × 5 km grid cell locations across Brazil and Mexico separately (Arita et al., 2008; Soberón & Ceballos, 2011). We identified a random sample of 5,000 rows from the PAM to choose random species combinations across Brazil and Mexico (separately) among which to test our classifiers based on real-world combinations of triatomine species (Fig. 3).

Figure 3 Map of species richness among the 39 triatomine species that are the focus of this analysis across Brazil, and 12 species across Mexico, each with three example sites and their corresponding triatomine faunas.

Distributional information is drawn from potential distributional estimates from ecological niche models.

Identifier testing

Given the limited number of samples (photos) for several of the species, we used a leave-one-out cross-validation approach for identifier testing, which maximizes the number of species that can be included. We use this testing approach as we are still in the process of building and testing the technology that comprises this tool. Eventually, once we create a complete reference set of images, all of the reference images can be used to create identifiers that can be used for any new test images.

To test identification abilities among species in local fauna sets, the input data are restricted to those species appearing in the local fauna, and training and testing of classifiers proceed as for the full set of species. To assess the related question of random species combinations (i.e., sets of species smaller than the whole 39- or 12-species set, but without the local-fauna constraint), we used the function “sample” in R to select random collections of specified numbers of species from the 39 Brazilian and 12 Mexican species to generate virtual species lists. For each number of species, we generated 200 such random faunas. For visualization of the spread of the different species in morphological spaces, we conducted a principal components analysis of the morphometric measurement matrix.

All of the program code and data on which this report is based is freely and openly available, in the hope that our work so far can provide a foundation of high-quality data inputs and progress-to-date in programming. Program code and the raw photographs of triatomines are available from the Dryad Digital Repository: http://dx.doi.org/10.5061/dryad.br14k; the data on which the final figure is based are available at KU Scholarworks (http://hdl.handle.net/1808/21560).

Results

We had available to us a total of 1,903 images of 67 Brazilian species and 428 images of 19 Mexican triatomine species (Table 1; see http://hdl.handle.net/1808/21560). Since few images were available for 28 of the Brazilian species and 7 of the Mexican species, efforts were concentrated on 39 Brazilian and 12 Mexican species for which sufficient images were available (12 in all cases, except Panstrongylus rufoturberculatus in Mexico, for which only seven were available). Automated morphological measurements from these images (N = 1,502 for Brazil and N = 396 for Mexico) and the landmarks extracted therefrom showed considerable spread in each country, suggesting that different species present measurably different shapes that can serve as an effective base for identification exercises (Fig. 4).

Figure 4 Illustration of the spread of different species of Brazilian and Mexican triatomine bugs in a morphological space defined by the first two principal components (Brazil: component 1 = 44.0% of overall variation, component 2 = 21.8% of overall variation; Mexico: component 2 = 18.7% of overall variation, component 3 = 12.5% of overall variation—note that species separated best in this space of component 3 vs component 2), summarizing all of the measurements used in this study.

(A) shows the distribution of all of the genera except for Triatoma, and (B) shows the distribution of Triatoma species only, as a zoom of the central part of (C).

In general, our classification efforts resulted in successful identifications of triatomine specimens. Even in the extreme challenge of identifying an image of a bug from among all species for which we had sufficient numbers of images, our average identification success rate (i.e., average of species averages) was 87.8% for Brazilian and 80.3% for Mexican species. We note that inclusion of two color characteristics improved the correct identification rate for T. pallidipennis from 69.8% to 90.7%. Indeed, this color characteristic alone improved overall correct identification rates even among the 39 Brazilian triatomines from 78.4% to 84.0%. Addition of the absolute-size characteristic (total length) provided further improvement of overall correct identification rates, yielding the final rates of 80.3% (Mexico) and 87.8% (Brazil).

The basic result is encouraging, as it indicates that sufficient information exists in the variation among species in shape to support automated identification, and that color- and size-related features may allow significant improvements. The identification ability rose notably when we moved to our reduced-fauna identification process (described below). Confusion matrices for each country as data sets (http://hdl.handle.net/1808/21560), which show how each image was classified correctly or incorrectly, and, in the latter case, for which species was the image mistaken; overall success rates are presented in Tables 1 and 2.

Looking across the 39 Brazilian species, identification success rates ranged from 100.0% (Eratyrus mucronatus, Panstrongylus diasi, Psammolestes tertius) to as low as 58.8% (T. williami). For seven species, identification success was 95% or above; for 18 species, success was 90% or above; and for 31 species, identification success was 80% or above. The worst-classified species (T. williami) was confused in three cases with T. matogrossensis, in two cases with T. guazu, (and others with which single images are confused); the second-worst-classified species (T. guazu) was mistaken for five different species. For Mexican species, identification success rates were lower, ranging from 100.0% (Panstrongylus rufotuberculatus) to as low as 60.0% (T. nitida); for three species, identification success was 90% or above; for seven species, identification success was 80% or above.

Exploring the effect of number of reference species on the success of identification exercises, we rarefied the species from each country at random to produce smaller numbers of species among which to discriminate (Fig. 5). For Brazilian species, the correct identification rate rose from 88.0% for 37-species subsets to 99.4% for 2-species subsets, in an approximately linear manner. This relationship translated into an improvement of 0.335% in classification rate for every reduction of one species. For Mexican species, identification success was uniformly lower (80.7% for 11 species, rising to 97.6% for 2-species subsets); curiously, the rate of improvement of identification success with smaller faunas was higher, at 2.11% per reduction of one species (Fig. 5).

Figure 5 Summary of classification success rates for 200 random combinations of numbers of species of 2–37 species for Brazilian triatomine faunas and 2–11 species for Mexican triatomine faunas (open symbols).

Also shown are success rates based on real-world species combinations at the testing sites (gray-filled symbols). Error bars are shown to indicate standard deviations for each fauna size.

A further question centered on the relationship between real-world sets of co-occurring species versus the random subsets analyzed. We used our library of maps of distributional areas to identify combinations of species potentially present at localities, which ranged as high as 12 species in Brazil and 11 species in Mexico (Fig. 3). Correct identification rates among the real-world sets of species did not differ significantly from the random reductions (Fig. 5): that is, the same relationship characterized real-world and random combinations.

Given the clear advantage of classifying images among smaller pools of species, the biggest challenge for our methodology is when the species image submitted is not among the pool among which one is testing; that is, when the presence of the species is a surprise (e.g., poorly known species’ distributions, human-mediated dispersal outside of the species’ distribution). On this front, we have encountered challenges; that is, as a backup, we can simply identify images against the broader suite of all 39 species in Brazil, for which we have a correct identification rate of 87.8%, though numbers for Mexico are a more disappointing 80.3% among 12 species. We conducted large numbers of tests, seeking some relationship between classifier outputs and whether or not the ‘true’ species is included in the pool, but have not encountered any clear signal.

Discussion

This study presents “full” automation of the identification process for the Triatominae, a medically important insect group. That is, with our system described above, once an image is captured, all processing is automated and no human intervention is needed. This level of autonomy distinguishes the Virtual Vector Lab from other such efforts (Mayo & Watson, 2007), and exists thanks to the multidisciplinary team that worked on this project from diverse perspectives: visual art, computer science, insect taxonomy, biogeography, etc. Some limitations still exist, and are reviewed in the next section.

Caveats

Although our overall, country-wide success rates were above 80%, our key to success in getting relatively high correct identification rates is the creation of small pools of candidate species from which to choose the answer, which we achieve via reference to distribution maps for each of the triatomine species. That is, we have dedicated considerable time and effort to the challenge of understanding geographic distributions of each triatomine species in Brazil (Costa, Peterson & Beard, 2002; Almeida et al., 2009; Gurgel-Gonçalves et al., 2011; Costa & Peterson, 2012; Gurgel-Gonçalves et al., 2012; Costa et al., 2014) and Mexico (Ramsey et al., 2000; Beard et al., 2002; Peterson et al., 2002; López-Cárdenas et al., 2005; Ibarra-Cerdeña et al., 2009; Ibarra-Cerdeña et al., 2014; Ramsey et al., 2015). These mapping efforts, of course, are based on what information is available, and—as such—can improve through time, particularly if the Virtual Vector Lab is eventually tied to databases that archive the results of large numbers of identification exercises for triatomines from many sites.

The dimension of the identification challenge in which further effort and innovation are needed is when the specimen to be identified is collected outside its predicted distribution and therefore is not represented in the local-fauna pool of species. We experimented at great length to address this problem, but were not successful in arriving at a concrete solution. Two interesting features that were noted were as follows. Results from pairwise species identification exercises were remarkably stable with regard to which species were included in the identification challenge. Our majority-vote approach to reconciling many individual species-by-species comparisons was usually won by a single vote—i.e., often, two species ‘tied’ as the most likely identification, so the winning vote was the direct, head-to-head comparison of those two species. Hence, it appears that little information is available to guide the identification process other than the head-to-head comparisons. Certainly, a good indicator of probable problems is when the local-fauna identification and the global-fauna identification do not agree; we have experimented with what to do when this “red flag” is raised, and have a number of possible solutions (e.g., add the global-fauna choice to the local fauna list, and re-run the estimator), but the success of this approach depends on the success rate of the global identifier. This challenge will be a focus of our continuing efforts.

Where our identification routines tend to fail more frequently is—not at all surprisingly—in comparisons within the species complexes that are known within the Triatominae (Lent & Wygodzinsky, 1979; Schofield & Galvão, 2009). For example, within the protracta complex in Mexico, our routines had great difficulty in differentiating T. barberi from T. nitida. Within the phyllosoma complex, six species are included in this analysis, and T. phyllosoma was the most often confused with complex-specific sister species, The three dimidiata complex haplogroups were as differentiated among themselves, as with all other species, the closest being the species morphologically most similar outside the complex (T. mexicana).

Future steps

For the moment, a constraint on the Virtual Vector Lab system is the requirement for high-quality images of the bugs. That is, much of our initial processing is possible thanks to the uniform blue background, the even lighting provided by our photo apparatus, and the use of a macro lens. We expect that combining our ongoing algorithm advancements with the rapidly increasing capabilities of cell-phone cameras should first allow discontinuation of need for a macro lens and dedicated iPod, and eventually of need for the photo platform apparatus itself, in successful analysis of triatomine images. Such developments would empower fully non-expert and community participation in development and use of the Virtual Vector Laboratory via any common cell phone and a dedicated application. Development and testing of those functionalities is a priority in the next phase of the project.

Another constraint on our present implementation is that of the initial assumption that the photograph is indeed a triatomine as a starting point to identification exercises. Distinguishing a triatomine from, say, a mosquito or a tick should be simple, but comparisons with other reduviids will be much more challenging. The reason that we chose triatomines as a first focal group for the Virtual Vector Lab is their size and relatively flat, two-dimensional morphology. To distinguish triatomines from other reduviids may require other views of the insect (antennae, legs, proboscis), although we have not as-yet conducted the necessary tests. Similar concerns apply to juvenile life stages: our present implementation is restricted to adults, the most likely stage in contact with humans. We may best deal with these challenges via instructions to users that will allow them to eliminate non-triatomines a priori, but that will also be explored in future phases of the project (Valdez-Tah et al., 2015).

A major source of optimism is that we have achieved very satisfactory correct identification rates working almost exclusively with information on shape of the bugs, which compares well with results of previous such efforts: 84% among five species of papilionid butterflies by Wang et al. (2012a), 93% among few species at the ordinal level by Wang et al. (2012b), 85.7–100% among 17 species of mosquitoes based on wing-shape characters (Lorenz, Ferraudo & Suesdek, 2015), and 90–98% among seven species of Neuroptera (Yang et al., 2015). From the two-dimensional images available to us, we can, in theory, take advantage of information on shape, size, and coloration. Using one of these three realms of information (shape), with some additional information from coloration and size, we have managed quite-useful accuracy in identifications, even among the three cryptic haplogroups traditionally considered under the name T. dimidiata (HG1 75.0%, HG2 80.0%, HG3 82.5%).

As an indication of the promise of still-better identification success that can come from adding information on color to our classifiers, we explored the utility of a classifier focused on the white patch on the corium of T. pallidipennis. Not only did this classifier resolve most of the confusion that our previous classifier had encountered with distinguishing between T. pallidipennis and T. phyllosoma (sister species), but indeed the addition of one color-related metric in our data matrix improved our classification success even for Brazilian triatomines by 3.2%. We obtained similar improvements in identification success upon including an absolute size measure. As such, in future efforts, we will explore more complete variable sets characterizing shape, size, and coloration.

Recently evolved species within complexes are clearly providing the greatest challenges for the system, even though the system is also giving us insights into shape and sister clades. The three T. dimidiata haplogroups in Mexico, clearly very closely related species, had previously been recognized based on molecular evidence, yet our morphometric analyses resulted in identifications with accuracy similar to that among other North American complexes. As such, we see great promise in a next generation of classifiers that will integrate morphometrics with characteristics corresponding to the interior of the body of the triatomines (as with the scutellum), and particular to the relative coloration of different portions of the body. We are also optimistic that specific classifiers and metrics can be built for each species pair that presents challenges to the more generic classifier that we have described in this paper.

Finally, extension of these general approaches to other groups is eminently feasible. Certainly, two-dimensional imagery is easiest to obtain and manage, such that organisms with relatively flattened body forms will be most tractable (e.g., ticks), although wings of other groups (e.g., mosquitoes and sandflies) may also offer opportunities (Zhou, Ling & Rohlf, 1985; Godoy et al., 2014; Lorenz, Ferraudo & Suesdek, 2015). Small size may present some level of challenge, as we will have to manage the complexities of magnification and associated distortion, but our photo apparatus has been designed to allow addition of accessories to facilitate such challenges in imaging. Ticks are probably the most logical next priority, in light of their size and relatively low species diversity (on the order of 700 species; Guglielmone et al., 2010), and considering the growing appreciation of ticks as significant vectors of many human and livestock diseases (Burgdorfer, 1977; Friedhoff, 1997; Parola, Paddock & Raoult, 2005). Agricultural pests will likely offer additional fruitful, tractable, and interesting sets of challenges.

In the future, we anticipate deployment of such systems more broadly—e.g., to medical personnel—as well as the broader public, to provide identification services for medically or economically important arthropod groups. The suite of methods described here provides a low-cost avenue to capture of high-quality images of triatomines and is promising to permit specific identification of triatomines. Quite simply, the broader initiative has the potential to open the knowledge of a handful of experts to a much-broader public that may benefit from access to entomological identification services.

We thank Lileia Diotaiuti (Fiocruz MG, Brazil), Mitermayer Reis (Fiocruz BA, Brazil), João Aristeu da Rosa (UNESP SP, Brazil), Renata Timbó (UnB, Brazil), Cleber Galvão and Jane Costa (Fiocruz RJ, Brazil), Eduardo Rebollar-Tellez (UANL, Mexico), Luis Alberto Hernandez-Osorio (UABJ, Mexico), and Jorge Lopez-Cardenas (LaESaP, Guanajuato, Mexico) for permitting photography of triatomine collections. We thank Renata Timbó (UnB, Brazil) for helping photograph triatomines at UnB. We thank Perry Alexander, Leonard Krishtalka, and Saralyn Reece-Hardy, for their leadership and guidance in assembling such an interdisciplinary team.

Additional Information and Declarations

Competing Interests

Author Contributions

Data Availability

The authors declare there are no competing interests.

Rodrigo Gurgel-Gonçalves and Janine M. Ramsey conceived and designed the experiments, performed the experiments, contributed reagents/materials/analysis tools, wrote the paper, reviewed drafts of the paper.

Ed Komp conceived and designed the experiments, performed the experiments, analyzed the data, wrote the paper, prepared figures and/or tables, reviewed drafts of the paper.

Lindsay P. Campbell and Hannah L. Owens performed the experiments, analyzed the data, reviewed drafts of the paper.

Ali Khalighifar performed the experiments, analyzed the data, prepared figures and/or tables, reviewed drafts of the paper.

Jarrett Mellenbruch conceived and designed the experiments, performed the experiments, reviewed drafts of the paper.

Vagner José Mendonça and Keynes de la Cruz Felix performed the experiments, contributed reagents/materials/analysis tools, reviewed drafts of the paper.

A. Townsend Peterson conceived and designed the experiments, wrote the paper, prepared figures and/or tables, reviewed drafts of the paper.

The following information was supplied regarding data availability:

Data sets: http://hdl.handle.net/1808/21560.

Images and code: http://dx.doi.org/10.5061/dryad.br14k.

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
