# Peer review of "Automated identification of insect vectors of Chagas disease in Brazil and Mexico: the Virtual Vector Lab"

_PeerJ, doi:10.7717/peerj.3040_

## Round 0.1 · original submission · Major Revisions

This is an interesting and potentially important development of image recognition software for automatic identification of disease vectors. Automated (and accurate) ID would provide important support to vector control professionals. However, while both reviewers recognized the potential of this work, both raised several shortcomings that need to be addressed before publication such as details regarding preparation and positioning of the specimen as well as access to more details of the code - after all, the idea is to provide this information as a starting point for others to develop. And there is a lot of room for improvement: (1) less "preparation" (i.e. ability to forego perfect positioning), (2) IDing of smaller specimens, (3) less reliance on geography. Indeed, if a human "handler" is required for the ID to be accurate then the value of the application is reduced significantly (it is relatively simple to train a technician to process and identify common specimens at a glance). However, the ability of a computer to compare the unknown specimens to hundreds or thousands of specimen information stored in a database would allow identification of "hard-to-ID" specimens (such as rare species or, more importantly, exotics). This would be an important achievement.

Reviewer 1 ·

Basic reporting

The article is both well written and comprehensive in presenting related information in the introduction and the discussion. The figures are designed and described very well. Raw data is supplied in the form of “confusion matrices”, a table of success rates of correct identification of photographs of various species and any misidentifications. No raw data is supplied for some more minor components of the paper such as the data presented in Figure 5 and the actual specimen photographs themselves and I think that these data may be beneficial to provide as well. Additionally, the code necessary for automating image processing, assessing various landmarks for identification and classifying an image to species, the main contribution of this paper, has not yet been provided.

Experimental design

The newly implemented automated procedure is well described with detailed and clear descriptions of how the estimator and classifier function. The testing of the classifier via the leave-one-out cross validation could be expounded upon slightly. In the future, do you imagine that the classifier will use every applicable photograph of every relevant species for classification potentially even incorporating new photographs of initially unknown specimens or a narrower set of training images?

Validity of the findings

The data is robust and conclusions are appropriate. The number of entries in the supplementary material (396 for Mexico; 1502 for Brazil) are slightly different from the number of photographs reported in the paper (412 for Mexico; 1504 for Brazil).

Additional comments

Gurgel-Gonçalves et al. demonstrate the first attempt at an automated process for identification of Triatominae species primarily through comparison of a set of estimated morphological landmark ratios estimated from a photograph to those of known species. The paper includes a component modeling species distributions for 17 species of Triatominae and is markedly more successful when photographs are classified only against a potential pool of species known to occur in a certain area. Though already mostly successful for the initial set of photographs, they provide details and ideas for future implementations to further increase the success of identification, e.g., by further inclusion of additional coloration characters for challenging species pairs. They adequately address caveats about their classifier and have thought about potential solutions for several problems. The only suggestions I have relate to inclusion of some additional data including the code required to implement the classifier itself, which would be crucial for another party to repeat the same experiment.

Line 43: Add comma after “Alternatives”

Reviewer 2 ·

Basic reporting

The submission is suitable to PeerJ policies.
Minor comments: The manuscrite include a good introduction and is appropriately referenced. One more reference should be referenced. There is a web and mobile tool to help taxonomists and health care technicians to overcome the difficulties to identify Triatomine. The software makes use of dichotomous key method and help the identification of triatomines species from Brazil. Freely available at http://triatokey.cpqrr.fiocruz.br/
Please, the text of lines 68 and 69 should be adapted.

Experimental design

Some doubts:
1) Line 117: The authors say that the pins were removed for capture images. But the figure 1 shows an insect on the pin. Is it necessary to use the pin or no? Both are possible? What is consequences to use specimen pin? Sometimes we cannot remove the pin of specimen voucher.
2) The authors say: “When possible, the first pair of legs is removed to simplify image processing” (line 120-122). But if it does not possible, what kind of injury can this bring to the analysis? Has any test been performed to determine the identification success rates under these conditions?
3) The magnification lens attached is always 7X? Even for large species like Panstrongylus megistus, Triatoma vitticeps, P. geniculatus, etc?
4) What can be doing to avoid problems in total length if the wings extend beyond the final tergits?
5) To avoid false positives, the technique depends on an accuracy of the landmarks position. How to ensure, for example, the same position vertically for all insects on photo apparatus?
6) How long does it take to process a insect and give the results? What is the value of apparatus? Finally, the automated system would be use by health care technicians to support routine entomological surveillance?

Others comments:
- The name of triatomine species must be written in accordance with the International Code of Zoological Nomenclature. Please, reviewer all the text and tables (e.g., lines 77, 281, 294).
- Table one available in Supplementary Materials is wrong. I did not see the data that shows a total of 1903 images of 67 Brazilian species and 428 images of Mexican triatomine species.
- The data which shows how each image was classified is wrong too. There is a lot of mistakes and the sample size presented in Tables 1 and 2 don’t match witch the Confusion matrices (Supplementary Materials).
e.g., See below (table 1)
Species sample size… …success rate
E. mucronatus 10 100
E. mucronatus 12 (according Confusion matrices) ??

Only values about P. geniculatus, R. montenegrensis, T. baratai, T. pintodiasi and T. mexicana vectors is correct. Please recalculate success rate according to correct sample size.
- Lines 392 - 400: to review (e.g., identification success rates was not 100.0% for P. diasi; T. williami was confused in 3 cases only with T. guazu; The sample size from P. rufotuberculatus is <9.
- After reassess of the data, the values should be changed in the text and tables.

Validity of the findings

The authors mentions and recognize some caveats, challenges, problems and constraint for the use of virtual vector Lab. Recognizing failures is a merit. But for me it was not clear the main conclusion of the work in the opinion of the authors.

Additional comments

This manuscript takes an interesting challenge, Namely a fully automated system that is able to identify triatomine bugs from Mexico and Brazil with a good accuracy. The question is very interesting, and potentially is very important. However, there is shortfalls need to be remedied before this manuscript can be considered as publishable.

---

## Round 0.2 · accepted · Accept

Thank you for your patience with the review of your manuscript. Evidently, the original editor is not available, and I have stepped in to move the manuscript toward publication. Both reviewers feel that you have adequately addressed their concerns and now recommend acceptance. I would just note that one reviewer had the following minor comments, that you may want to address in the final copy of your publication (I will alert PeerJ in case you would like to make these corrections and submit a final version).

Line 50: While 150+ species is technically correct, including the 2 fossil species, perhaps "149 extant species" or nearly "150 extant species" would be more accurate in this instance
Line 120: Although I can not tell for certain, the pin seems to be in the thorax in between the legs and not the abdomen, which would be an unusual and unstable place for mounting a specimen.
Line 225 and rest of paragraph: "tergits" should be "tergites"

Reviewer 1 ·

Basic reporting

The authors have deposited the data I had previously requested and I have no further comments.

Experimental design

The authors have addressed the small question I had previously requested and I have no further comments.

Validity of the findings

The authors have corrected the inconsistency in their data and explained how it occurred and I have no further comments.

Additional comments

In looking at the new changes, I have a three minor things I overlooked at first or were newly added.

Line 50: While 150+ species is technically correct, including the 2 fossil species, perhaps "149 extant species" or nearly "150 extant species" would be more accurate in this instance
Line 120: Although I can not tell for certain, the pin seems to be in the thorax in between the legs and not the abdomen, which would be an unusual and unstable place for mounting a specimen.
Line 225 and rest of paragraph: "tergits" should be "tergites"

Reviewer 3 ·

Basic reporting

This is not a full review, just a quick response as requested by the handling editor.

Experimental design

The design is very much based on trial and error. While the process and reasoning for the trials are well described, others attempting to use or expand on the methods will probably face their own, different, headaches and have to go through their own process. To that end the specific methods are unlikely to be replicable en masse, though some may prove useful to others.

Validity of the findings

As a lot of the paper involves trial and error the results--the reported success rates are contingent on the numerous trials and errors. For instance the accuracy of one datum included depends 'on the accuracy with which the legs have been clipped off in earlier steps', the success rates presented in turn depend (or might, the neural net is a bit of a black box) on that datum, which means in term these success rates depend on how you clip of the legs. And how you rescale the background. And the micro-decisions of Garp, and dozens of other things. This is not to say the results are not 'valid', but it is to say they are only valid in the context in which the work was undertaken.